# Mitigation of China's carbon neutrality to global warming

Longhui Li [1,2,3], Yue Zhang[1,2,3], Tianjun Zhou [4], Kaicun Wang [5], Can Wang [6], Tao Wang[7], Linwang Yuan[1,2,3], Kangxin An[6], Chenghu Zhou[8] & Guonian Lü[1,2,3] ✉

Projecting mitigations of carbon neutrality from individual countries in relation to future global warming is of great importance for depicting national climate responsibility but is poorly quantified. Here, we show that China's carbon neutrality (CNCN) can individually mitigate global warming by 0.48 °C and 0.40 °C, which account for 14% and 9% of the global warming over the long term under the shared socioeconomic pathway (SSP) 3-7.0 and 5-8.5 scenarios, respectively. Further incorporating changes in $CH_4$ and $N_2O$ emissions in association with CNCN together will alleviate global warming by 0.21 °C and 0.32 °C for SSP1-2.6 and SSP2-4.5 over the long term, and even by 0.18 °C for SSP2-4.5 over the mid-term, but no significant impacts are shown for all SSPs in the near term. Divergent responses in alleviated warming are seen at regional scales. The results provide a useful reference for the global stocktake, which assesses the collective progress towards the climate goals of the Paris Agreement.

Global warming since the preindustrial era has been primarily attributed to the increase in atmospheric $CO_2$ concentrations, which mainly results from the carbon emissions of fossil fuel combustion[1,2]. The likely range of the total human-caused global surface temperature increase from 1850–1900 to 2010–2019 is 0.8 °C to 1.3 °C, with a best estimate of 1.07 °C[2]. The contributions of historical anthropogenic carbon emissions have been quantified, although the currently existing qualifications are based on different criteria and differ in estimation[3,4]. If worldwide carbon emissions continue at the current rate, global warming is likely to exceed 1.5 °C between 2030 and 2052, and even more than 3–5 °C at the end of the 21st century[2]. Global warming has caused a range of broad threats to both natural system and humanity, including increasing extreme climate events, rising sea levels, and shifting wildlife populations and habitats[5]. To limit the increase in global mean temperature below 1.5 °C above preindustrial levels, reaching net zero of global $CO_2$ emissions in 2055 and limiting non-

$CO_2$ greenhouse gas (GHG) emissions after 2030 are crucial mitigation strategies[6]. Since the Intended Nationally Determined Contributions to mitigating global warming agreed upon in the 2015 Paris Agreement, more than 120 countries have pledged to achieve carbon neutrality and declared a date[7]. In the context of this unique opportunity, projecting mitigations of pledged carbon neutrality to future global warming is of the same importance as previous efforts devoted to quantifying historical climate responsibilities, which can inform the implementation of global climate mitigation strategies and equality in development from nations' carbon emissions. As one of the large annual emitters of $CO_2$ emissions[8], China has committed to peak its carbon emissions before 2030 and attain carbon neutrality before 2060[7,9]. Such an ambitious policy target for $CO_2$ emission reduction is expected to mitigate global warming. A recent study based on a very simplified climate model reported that China's carbon neutrality alone will contribute a 0.16–0.21 °C avoided global warming at the end of the

[1]Jiangsu Center for Collaborative Innovation in Geographical Information Resource Development and Application, Nanjing 210023, China. [2]Key Laboratory of Virtual Geographic Environment (Nanjing Normal University), Ministry of Education, Nanjing 210023, China. [3]School of Geographical Sciences, Nanjing Normal University, Nanjing 210023, China. [4]LASG, Institute of Atmospheric Physics, Chinese Academy of Sciences, Beijing, China. [5]Sino-French Institute for Earth System Science, College of Urban and Environmental Sciences, Peking University, Beijing 100871, China. [6]School of Environment, Tsinghua University, Beijing, China. [7]State Key Laboratory of Tibetan Plateau Earth System and Resources Environment (TPESRE), Institute of Tibetan Plateau Research, Chinese Academy of Sciences, Beijing, China. [8]Institute of Geographical Information Science and Natural Resources, Chinese Academy of Science, Beijing, China. ✉e-mail: gnlu@njnu.edu.cn

21st century[10]. However, the magnitude of such mitigation has not yet been quantified using a fully coupled Earth system model that incorporates all crucial components of the climate system.

In this work, we employ the NCAR Community Earth System Model (CESM)[11] to project the mitigation of China's carbon neutrality (CNCN) to global warming by comparing four pairs of simulations corresponding to four shared socioeconomic pathways (SSPs), including SSP1-2.6, SSP2-4.5, SSP3-7.0 and SSP5-8.5, which represent different anthropogenic surface $CO_2$ emissions in future scenarios of socioeconomic development, population growth, and land and water requirements for food supplies[12]. Each pair of simulations consists of one default from the sixth phase of Coupled Model Intercomparison Project (CMIP6) built-in CESM and another one in which anthropogenic surface $CO_2$ emissions in China's domain are replaced with values in the policy target of CNCN released by Tsinghua University in 2021[13] (Supplementary Fig. S1). The results show that China's carbon neutrality (CNCN) can individually mitigate global warming by 0.48 °C and 0.40 °C, which account for 14% and 9% of the average increase in global mean surface temperature (GMST) over the long term (2081–2100) under the SSP3-7.0 and SSP5-8.5 scenarios, respectively. When further incorporating changes in $CH_4$ and $N_2O$ emissions in association with CNCN (CNCN_ext), the combined effects of three primary GHGs will slow down the GMST by 0.21 °C and 0.32 °C for SSP1-2.6 and SSP2-4.5 over the long term, and even by 0.18 °C for SSP2-4.5 over the mid-term (2041–2060).

## Results

### Temporal evolutions of mitigations of China's carbon neutrality to global warming

Following the IPCC AR6[2], we examine the projected changes in GMST for the near term (2021–2040), mid-term (2041–2060) and long term (2081–2100) relative to 1850–1900 (preindustrial period). Compared to the preindustrial period, the GMST averaged over 2081–2100 is projected to increase by 1.7 °C under the low GHG emissions scenario (SSP1-2.6), by 2.7 °C and 3.4 °C under the two intermediate scenarios (SSP2-4.5 and SSP3-7.0), and by 4.7 °C under the very high GHG emissions scenario (SSP5-8.5) (Fig. 1a). The simulated GMSTs for the policy target of the CNCN scenario do not significantly differ from those for the corresponding SSPs over the near term (Fig. 1b), and the insignificant response in the near-term projection is caused mainly by the CESM's internal variability[14]. Such insignificant mitigation is also the case over the mid-term except for SSP5-8.5, in which CNCN mitigates the impact of global warming by 0.17 °C (±0.05 °C) (Fig. 1c). Over the long term, the GMST in the policy target of CNCN scenarios is significantly different ($p < 0.01$) from that in the default SSP scenarios, except for SSP1-2.6 (Fig. 1d). For SSP2-4.5, China's carbon neutrality

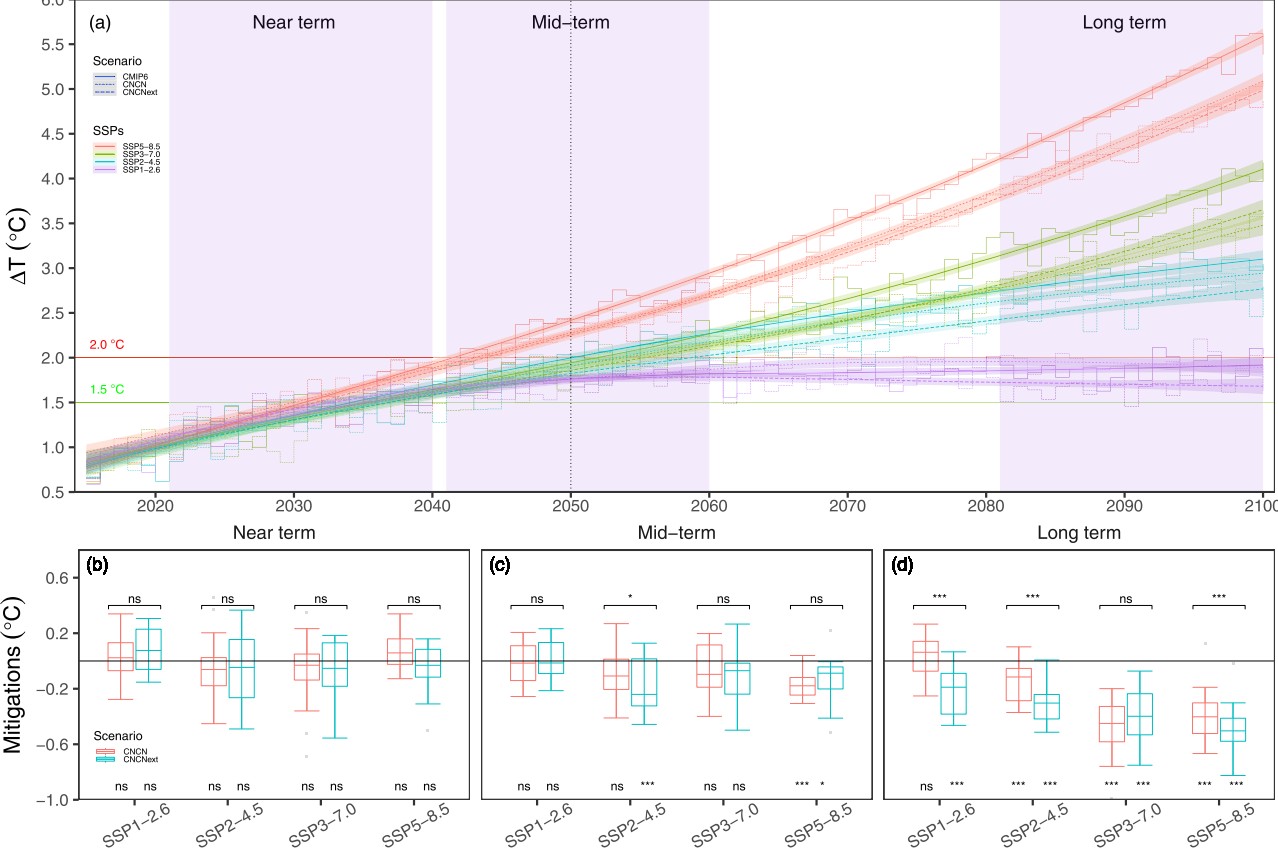

**Fig. 1 | Mitigations of China's carbon neutrality (CNCN) and extension scenarios with additional $CH_4$ and $NO_2$ emission reductions accompanied by the $CO_2$ emission reductions for CNCN (CNCN_ext) to global mean surface temperature (GMST). a** Evolution of GMST changes relative to the preindustrial period (1850–1900 average) (ΔT, unit in °C) for the default CMIP6, CNCN and CNCN_ext scenarios from 2015 to 2100. **b–d** represent mitigations of the CNCN and CNCN_ext scenarios for each shared socioeconomic pathway (SSP) over the near term (2021–2040), mid-term (2041–2060) and long term (2081–2100). The mitigation effect is calculated as the difference in GMST between the CNCN (or CNCN_ext) and the default CMIP6 scenarios. The segment lines represent the original values and the shaded areas around the smooth line represent the confidence at level of 0.05 for each scenario. The box plots in panels **b–d** show the 25th, the median and the 75th percentile range (box) and the minimum–maximum range (whiskers). The symbols * and *** represent statistical significance at levels of 0.01 and 0.001, respectively. "ns" represents statistical insignificance at level of 0.01. Symbols with underscores shown above the head of the box in **b–d** represent comparisons between the CNCN and CNCN_ext scenarios and those shown above the x-axis are referred to as comparisons between the CNCN (or CNCN_ext) and the default CMIP6 scenarios. The paired t test was used to calculate the significance.

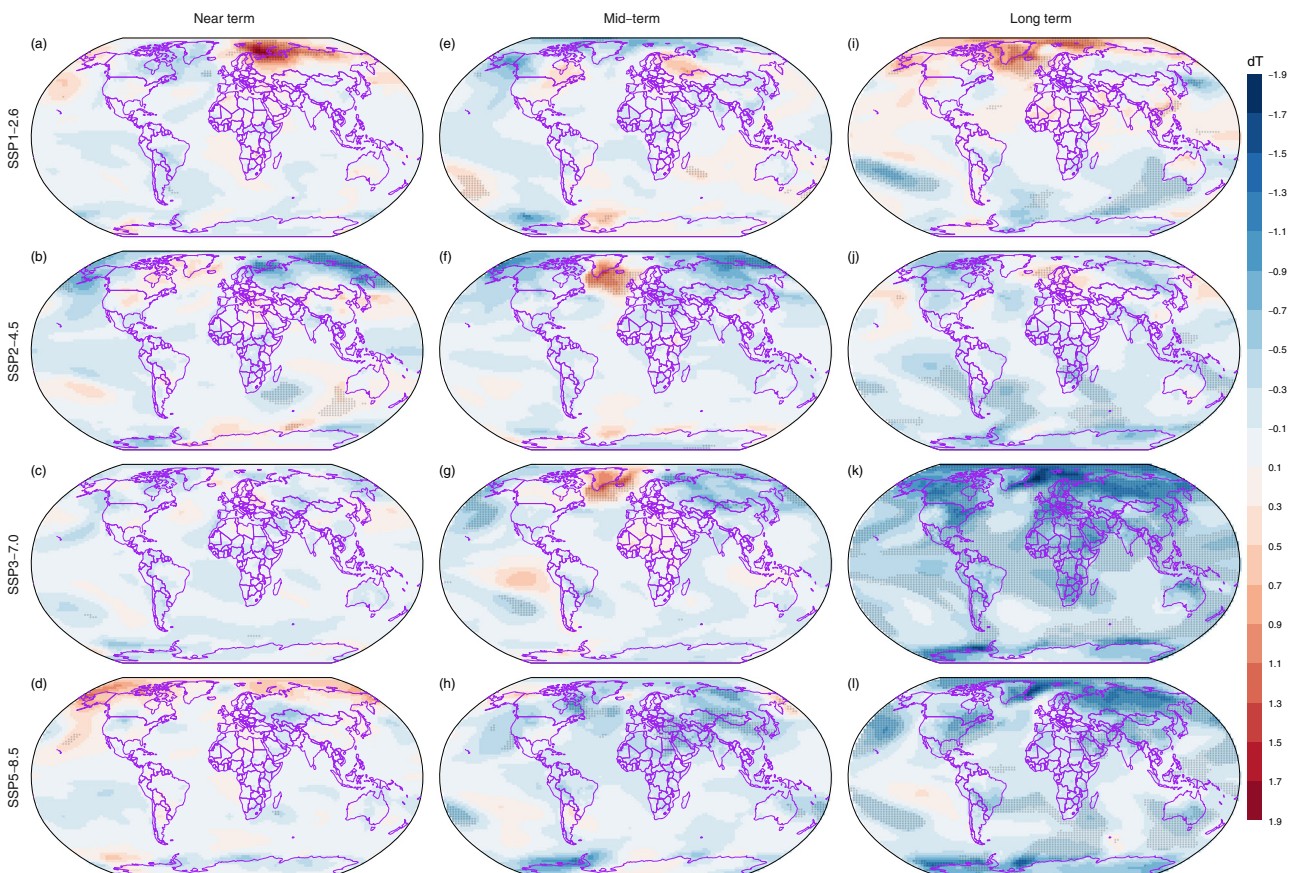

**Fig. 2 | Mean surface temperature difference between China's carbon neutrality (CNCN) and default CMIP6 scenarios for four shared socioeconomic pathways (SSPs).** Panels from left to right represent the near term (2021–2040, **a**–**d**), mid-term (2041–2060, **e**–**h**) and long term (2081–2100, **i**–**l**), and panels from top to bottom represent SSP1-2.6, SSP2-4.5, SSP3-7.0 and SSP5-8.5, respectively. The mean surface temperature difference (dT, unit in °C) is calculated as the value of CNCN minus the default CMIP6 for each combination of the study term and SSPs. Pixels overlayed by dots indicate that the dT is statistically significant at a level of 0.01. The paired *t* test was used to calculate the significance.

reduces the GMST by 0.14 °C (±0.07 °C) in the long term. For the SSP3-7.0 and SSP5-8.5 scenarios, CNCN potentially reduces the long-term GMST by 0.48 (±0.09) °C and 0.40 (±0.09) °C ($p < 0.01$), respectively (Fig. 1d). Such mitigations account for 14% and 9% of the average increase in the GMST during the last two decades of the 21st century, respectively.

When further incorporating the changes in $CH_4$ and $N_2O$ accompanied by the reductions in carbon emissions (i.e., CNCN_ext), the CNCN_ext scenario does not result in significant changes in GMST for all four SSPs over the near and mid-terms except for SSP2-4.5, which shows ~0.09 °C less warming over the mid-term ($p < 0.01$) compared with the CNCN scenario (Fig. 1b, c). Over the long term, changes in $CH_4$ and $N_2O$ emissions prevent significant further GMST warming relative to the CNCN scenario for all SSPs except for SSP3-7.0 (Fig. 1d). Consequently, changes in the three GHG emission under the pledge of China's carbon neutrality result in significant and large net declines in GMST by 0.21 (±0.17), 0.32 (±0.13), 0.50 (±0.21) and 0.39 (±0.17) °C over the long term for SSP1-2.6, SSP2-4.5, SSP3-7.0 and SSP5-8.5, respectively (Fig. 1d). Such impacts on the GMST are significant for SSP2-4.5 (0.18 ± 0.09 °C) and SSP5-8.5 (0.13 ± 0.07 °C) over the mid-term but insignificant for all SSPs over the near term. For those pairs of simulated GMST without significant differences, underlying causes mainly come from either smaller differences among the default CMIP6, CNCN and CNCN_ext scenarios (Supplementary Figs. S3, S4) or historical $CO_2$ accumulation up to the simulation years[4], as well as the internal variability[14]. As previously reported, the internal variability of the CESM ranges from 0.06 to 0.09 °C[15], and the magnitudes of mitigations on global warming from CNCN and CNCN_ext for the SSP3-7.0 (0.48 and

0.39 °C) and SSP5-8.5 (0.4 and 0.5 °C) scenarios over the long term are larger than the internal variability, indicating a robust response.

## Spatial divergences of mitigations of China's carbon neutrality to global warming

The effect of anthropogenic $CO_2$ emissions on GMST is evident at both global and regional or local scales[16]. The change of surface temperature induced by CNCN (dT) is scenario-dependent, spatially not uniform and varies with time. The magnitudes of dT at the global scale range from −1.84 to 1.76 °C (Fig. 2). Remarkable differences are seen at regional scale. Over the near term and mid-terms, only 0.2–4% of grids show significant differences in dT at the global scale (Fig. 2a–d). A strong warming is seen in the near term under SSP1-2.6 over the Barents Sea and its southern regions adjacent to western Russia (Fig. 2a). Such kind of warming can be explained by the fact that the SSP1-2.6 under the CNCN scenario has significantly larger carbon emissions than the default scenario over the near term (Supplementary Fig. S2). Over the mid-term, CNCN leads to a strong increase in surface temperature in south of Greenland under both the SSP2-4.5 and SSP3-7.0 scenarios but a significant decrease in east of Siberian sea and adjacent to eastern Russia under the SSP2-4.5 scenario (Fig. 2f, g), which is highly consistent with the report of a significant regional transient response of local temperature to $CO_2$ emissions[16]. Over the long term, grids with significant differences in dT account for up to 10% and 9% of the global values for SSP1-2.6 and SSP2-4.5, respectively, but the signs of CNCN-induced surface temperature change are regionally dependent (Fig. 2i–l). Warming effects over the long term in Greenland and the Arctic from SSP1-2.6 are primarily caused by higher

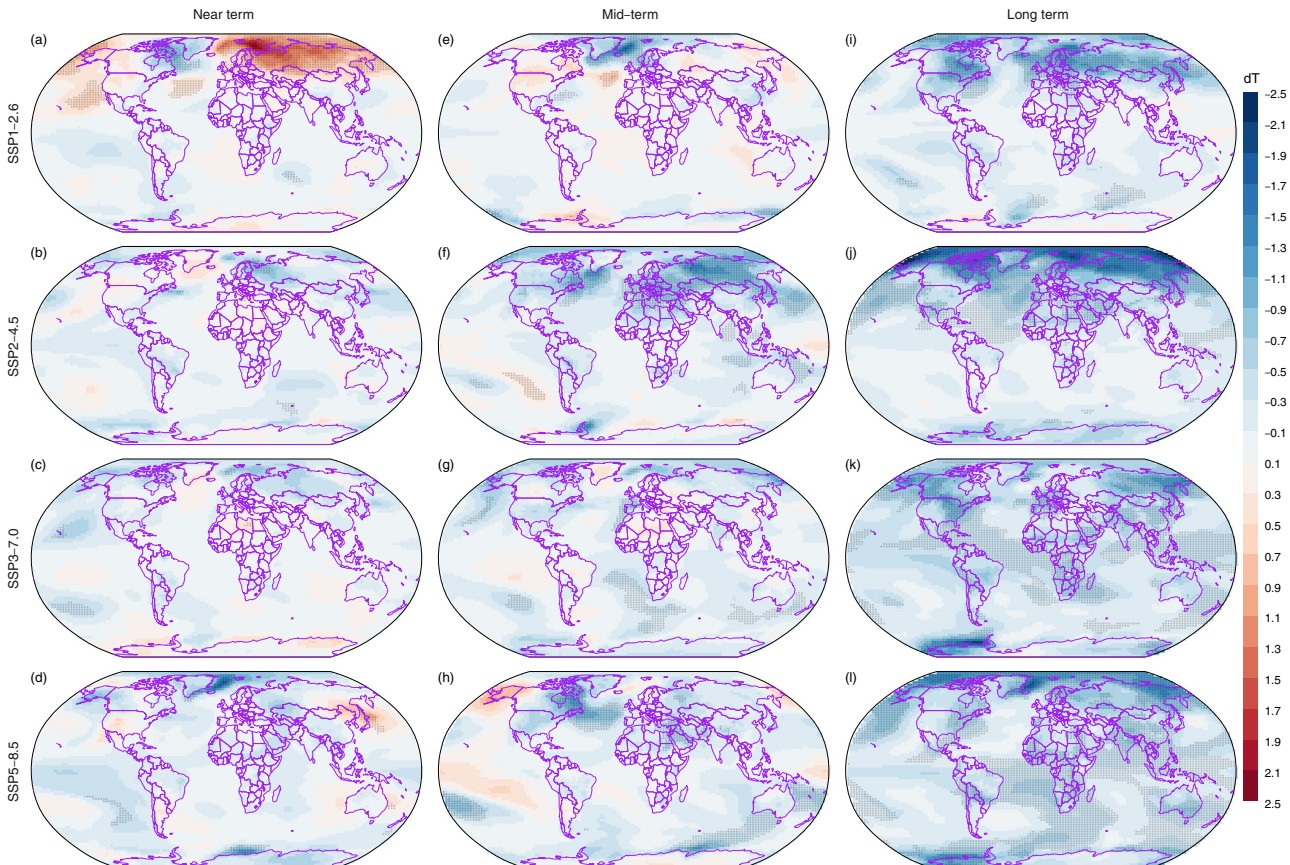

**Fig. 3 | Mean surface temperature difference between extension scenarios with additional CH$_4$ and NO$_2$ emission reductions accompanied by the CO$_2$ emission reductions for China's carbon neutrality (CNCN$_{ext}$) and default CMIP6 scenarios for four shared socioeconomic pathways (SSPs).** Panels from left to right represent the near term (2021–2040, **a–d**), mid-term (2041–2060, **e–h**) and long term (2081–2100, **i–l**), and panels from top to bottom represent SSP1-2.6, SSP2-4.5, SSP3-7.0 and SSP5-8.5, respectively. The mean surface temperature difference (dT, unit in °C) is calculated as the value of CNCN$_{ext}$ minus the default CMIP6 for each combination of the study term and SSPs. Pixels overlayed by dots indicate that the dT is statistically significant at a level of 0.01. The paired t test was used to calculate the significance.

anthropogenic CO$_2$ emissions than the default scenario (Supplementary Fig. S2). The avoided warming from SSP2-4.5 is mainly located in the Southern Ocean (Fig. 2j). For SSP3-7.0 and SSP5-8.5, 53% and 34% of the global values show significant differences in dT, respectively (Fig. 2k–l). Generally, the avoided warming induced by CNCN over the long term under SSP3-7.0 scenario features a polar amplification pattern, with stronger avoided warming in the Arctic and high-latitudes, and weak avoided warming in low latitudes (Fig. 2k). Under the SSP5-8.5 scenario, strong avoided warming is seen in eastern Greenland, the Greenland Sea, large regions of Russia and some fractions of regions of land and oceans from low to high latitudes (Fig. 2l).

Compared to the CNCN scenario, the joint effects of CH$_4$ and N$_2$O further result in additional near-term warming under both SSP1-2.6 and SSP2-4.5 scenarios but significantly less warming under SSP5-8.5 scenario in high latitudes of the Northern Hemisphere (Supplementary Fig. S7), which is mainly due to the difference in N$_2$O emissions between the two scenarios (Supplementary Fig. S4). Further avoided warmings from the combined effects of CH$_4$ and N$_2$O are more evident under both SSP1-2.6 and SSP2-4.5 scenarios in the long term in high latitudes of the Northern Hemisphere. However, significant warming induced by CH$_4$ and N$_2$O emissions is observed over the long term under SSP3-7.0 scenario in the regions surrounding the Mediterranean Sea and the Barents Sea (Supplementary Fig. S7). Taking comprehensive covariations of three primary GHGs within the China's carbon neutrality pledge into account results in a strong warming effect in near-term GMST over a large area covering the majority of the Barents Sea, Russia and its

northern oceans over the near term for SSP1-2.6 (Fig. 3a). The response of surface temperature in the near-term is not evident under all other scenarios (Fig. 3a–d). The avoided warming in the mid-term under CNCN$_{ext}$ scenario is only evident in small regions, which covers only 3% (SSP1-2.6) to 10% (SSP2-4.5) of global area (Fig. 3e, f). In contrast, a remarkable avoided warming is seen in the long term, which accounts for 8%, 22%, 25% and 36% of the global area for SSP1-2.6, SSP2-4.5, SSP3-7.0 and SSP5-8.5, respectively (Fig. 3i–l).

## Discussion

Here, we make a first attempt to project the effect of pledged carbon neutrality from an individual country, taking the currently largest emitting country as an example, on future global warming mitigation based on a fully coupled Earth system model. The avoided warmings from CNCN are significant under SSP3-7.0 and SSP5-8.5 scenarios over the long term but insignificant or significant in only less than 10% of global regions under other scenarios for near term and mid-term. Mitigations caused by CNCN alone would reduce global warming by 0.48 (±0.09) °C and 0.40 (±0.09) °C in these two scenarios, respectively. Such mitigation magnitudes account for 14% and 9% of global warming in the same period. When further changes in CH$_4$ and N$_2$O emissions in association with China's carbon neutrality are incorporated, the combined effects of the three primary GHGs slowed the warming of GMST in the long term by 0.21–0.50 °C for the four SSPs, which accounted for approximately 11–12% of global warming. CNCN$_{ext}$ also contribute avoided warmings for SSP2-4.5 (0.18 °C) and

SSP5-8.5 (0.12 °C) over the mid-term but does not have any significant impacts on the GMST for all four SSPs in the near term.

We also acknowledge the uncertainties in our study. First, in our numerical experiments, we assumed that other countries do not take any significant mitigation actions, which is far different from future reality, as more than 120 countries have pledged such actions[7]. Given that many efforts to reduce GHG emissions worldwide have been pledged, China's share of the mitigation of global warming will largely decline. Second, the emission pathway used to drive the Earth system model in this study is only one of various possible pathways to attain carbon neutrality, and emission pathways for carbon neutrality are quite sensitive to variations in nations' policy interventions, the decarbonization progress in electricity and industrial sectors, and establishment of renewable energy and technology innovations for carbon capture and storage or carbon dioxide removal[17]. Third, changes in atmospheric aerosols, such as short lived GHGs cooccurring with fossil fuel combustions and diverse human activities, are not taken into account in the simulations, although the combined effect of various kinds of aerosols is reported to have conflicting effects that range from significant reduction in temperature to a modest impact and even a net future warming effect[18–21]. Finally, mitigation effects are derived by pair simulations of a single factor, $CO_2$ only or a combination of $CO_2$, $CH_4$ and $N_2O$, which essentially cannot sort out an individual country's effects on global warming because the global warming is caused by the cumulative anthropogenic $CO_2$ emissions from the preindustrial era and the historical emission will still work in future warming. In addition, multiple simulations for the same scenario would reduce the uncertainty induced by the model's internal variability and make the results more robust.

It is well known that $CO_2$ emission pathways for carbon neutrality and GHG neutrality are different. Carbon neutrality targets a balance between anthropogenic emissions by sources and removals by sinks of carbon, but GHG neutrality refers to all greenhouse gases, which means that additional negative $CO_2$ emissions and some non-$CO_2$ GHG emissions have to cancel out each other for GHG neutrality[22–24]. China has delivered a series of domestic strategies and policies including abated coal consumption[25], clear energy development[26–28], nationwide ecological restoration[29,30] and other various negative-emission technologies[31] as potential countermeasures to achieve carbon neutrality by 2060. Most of these China's ongoing emission actions also contribute to reductions in some non-$CO_2$ GHG emissions and increases in negative-emission of $CO_2$, which implies that China's future emission pathway is ultimately targeting for a GHG neutrality, although carbon neutrality is currently claimed[7,9]. China's rapid growth in anthropogenic $CO_2$ emissions occurred after 2000 (Supplementary Fig. S2), and the time spans to carbon peaking (2030) and neutrality (2060) are quite short, compared to those of developed countries[4]. This creates great challenges for China to achieve carbon or GHG neutrality. However, China's determination and ambition for carbon or GHG neutrality are very strong because impelling for carbon and GHG neutrality will not only contribute to mitigating further global warming but also to facilitating economic transformation and upgrade for more sustainable development, including improving domestic air quality and protecting public health[32].

This study quantifies the relative contribution of China's carbon neutrality pledge to future global warming, which is informative and insightful for quantifying an individual country's mitigation of carbon emissions on scales of both time horizons and spatial distributions. Our results provide a useful reference for the global stocktake, which assesses the collective progress towards the climate goals of the Paris Agreement. This also implies that joint efforts from all countries in the world are urgently needed to mitigate further global warming through net-zero carbon actions.

## Methods
### Communication earth system model
The NCAR's Community Earth System Model (CESM)[11] is a fully coupled earth system model consisting of seven prognostic components, namely atmosphere, land, land ice, ocean, sea ice, river and a coupler that computes fluxes between components. The CESM 2.1.3 version incorporates a suite of cases for CMIP6 simulations with several shared socioeconomic pathway (SSP) scenarios. Four SSPs were used in this study. The first is SSP1-2.6, which represents the low end of the range of future forcing pathways and updates the RCP2.6 (RCP, representative concentration pathways). The second is SSP2-4.5, which represents the medium part of the range of future forcing pathways and updates the RCP4.5 pathway. The third is SSP3-7.0, which represents the medium to high end of the range of future forcing pathways. The fourth is SSP5-8.5, which represents the high end of the range of future pathways and updates the RCP8.5[12], although the SSP5-8.5 scenario is criticised as overestimating future cumulative fossil fuel and industry $CO_2$ emissions[33]. Each default CMIP6 SSP incorporates corresponding spatially anthropogenic surface $CO_2$ emissions produced with the Integrated Assessment Model (IAM).

### Global $CO_2$ forcing data under CNCN scenario
The CNCN scenario[13] is mainly based on carbon emissions consistent with the IPCC 1.5 °C target, but it requires further reductions in national total energy consumption and large increases in the proportion of nonfossil energy in primary energy consumption. The CNCN scenario also requires significant decreases in non-$CO_2$ GHG emissions and increases in terrestrial ecosystem carbon sinks and large-scale implementations of carbon capture and storage (CCS) and carbon dioxide removal (CDR). Unlike IAMs, it is not possible to generate spatially explicit anthropogenic surface $CO_2$ emissions for the CNCN scenario, but a roadmap with the expected amount of totally domestic $CO_2$ emissions for CNCN up to 2050 is generated[13]. We assume that future anthropogenic $CO_2$ emissions for each grid under the CNCN ($CO_2(i, j)$) scenario are linearly proportional to their original values in the default SSPs ($CO_{2,SSP}(i, j)$), i.e., the Eq. (1).

$$CO_2(i, j) = \frac{CO_{2,CNN}}{\sum_1^n CO_{2,SSP}(i, j)} \times CO_{2,SSP}(i, j) \qquad (1)$$

Generated spatial $CO_2$ emissions then replace the values in default SSPs in China but remain unchanged outside of China, and anthropogenic $CO_2$ emissions for the CNCN scenario after 2050 remain unchanged as its value in 2050. The new datasets of $CO_2$ emissions for CNCN are used to drive the CESM and represent the CNCN scenarios. Therefore, four pairs of simulations are formed, and the difference in each pair of simulations represents the mitigations of global warming from CNCN.

From the CNCN report[13], China's anthropogenic carbon emissions will peak at 10.5 $GtCO_2$ year$^{-1}$ in 2030 and drop to 1.2 $GtCO_2$ year$^{-1}$ in 2050. Under the carbon neutrality pathway[13], China will reduce its carbon emissions by 89% in 2050, which is roughly consistent with a recent synthesis for the 1.5 °C target based on multiple integrated assessment models[17]. Compared with the default CMIP6 scenario, CNCN has a difference ranging from −3.70 to 18.03 $GtCO_2$ year$^{-1}$ in anthropogenic surface $CO_2$ (Supplementary Figs. S2–S3).

### Global $CH_4$ and $N_2O$ forcing data under CNCN scenario
Because both $CH_4$ and $N_2O$, two other kinds of greenhouse gases, are emitted along with the fossil fuel production, transportation or combustion and other anthropogenic activities[9,34], changes in $CH_4$ and $N_2O$ concomitant with CNCN are also incorporated to drive the CESM (referred to as CNCN$_{ext}$). Under the CNCN$_{ext}$ scenario, China's cumulative $CH_4$ emissions during the period 2015-2100 have differences of −785, 810, 3552, and 661 Mt $CH_4$ compared to the default SSP1-2.6,

SSP2-4.5, SSP3-7.0 and SSP5-8.5, respectively. In contrast, the $CNCN_{ext}$ scenario results in changes in cumulative $N_2O$ emissions of −19.4, 8.2, 24.1 and 30.4 Mt $N_2O$ compared to the four default SSPs, respectively.

Both $CH_4$ and $N_2O$ forcing data of the CESM are prescribed as globally uniform surface concentrations, rather than their surface fluxes. Conversion from fluxes to concentrations generally requires running a simple climate model like MAGICC[35]. Under scenarios of four default SSPs, concentrations of $CH_4$ and $N_2O$ can be easily extracted. However, reimplementing an MAGICC for the CNCN scenario essentially involves a large number of simulation tasks. Fortunately, dependences of both $CH_4$ and $N_2O$ concentrations on their cumulative emissions in the troposphere during the period 2015-2100 can be empirically fitted by cubic functions for four SSPs under the default CMIP6 scenario ($R^2$ close to 1, Supplementary Fig. S5), although their fitting equations are largely different among variables and SSPs, so we have high confidence in deriving corresponding surface concentrations of $CH_4$ and $N_2O$ based on their cumulative emissions during a specified period (2015–2100) (Supplementary Fig. S5). As China has yet specified non-$CO_2$ GHG emission reduction targets and timelines for GHG neutrality, the Global Change Assessment Model[36] (GCAM, version 5.4) is used to project emissions of $CH_4$ and $N_2O$ under the CNCN scenario from 2015 to 2100, in which emissions of $CO_2$ under the CNCN scenario are used as constraints, and an equal marginal cost of emission reductions between $CO_2$ and non-$CO_2$ GHGs is assumed. Therefore, concentrations of $CH_4$ and $N_2O$ can be derived under both scenarios of the default CMIP6 and CNCN (Supplementary Fig. S6). Such CESM simulations driven by combined variations in $CO_2$, $CH_4$ and $N_2O$ under the CNCN scenario represent the $CNCN_{ext}$ scenario, whose differences in simulated GSMT from CNCN simulations are regarded as relative contributions to global warming by both $CH_4$ and $N_2O$ but those differences in simulated GMST from the default CMIP6 simulations are regarded as contributions by $CNCN_{ext}$.

### Prior validation of the community earth system model
Prior to beginning simulations, the CESM 2.1.3 was run with fully coupled components from both 1850 and 1900 to 2014, and no significant difference in the simulated GMST was found (Supplementary Fig. S8), which allowed us to use the spin-up simulation from 1850 to 2014 for the start-up simulation. The simulated GMST from spin-up runs was also compared with three global historical temperature datasets, and the consistency identified by cross-corelation was high (R ranging from 0.78 to 0.99) (Supplementary Fig. S9), proving the robustness of the model in simulating the response of global mean temperature to anthropogenic $CO_2$ emissions.

## Data availability
Dataset of future $CH_4$ and $N_2O$ emissions for different SSPs are archived in IIASA website (https://tntcat.iiasa.ac.at/SspDb). Due to the extremely large size of the output raw data from CESM simulations, these files were not deposited in a public repository, but are available from the corresponding authors on reasonable request.

## Code availability
The R codes for visualizing the results in this study are available from the corresponding authors on request.

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

## Acknowledgements

This study is supported by the National Key Research and Development Program of China (2017YFA0603603) and the National Natural Science Foundation of China (U2003201).

## Author contributions

L.L. and G.L. designed research and drafted the paper; L.L., Y.Z., and K.A. performed the numerical simulations; L.L., Y.Z., and T.Z. analyzed the results; and L.L., T.Z., K.W., C.W., T.W, L.Y., and C.Z. revised the paper. L.L., Y.Z., T.Z., K.W., C.W., T.W., L.Y., C.Z., and G.L. contributed to the interpretation of the results and to the text.

## Competing interests

The authors declare no competing interests.
