## [Peer Review File · Nature Communications]

Reviewer comments, first round review

Reviewer #1 (Remarks to the Author):

The paper under review provides quantitative assessment of potential contributions of carbon neutrality of Chinese economy to global and regional temperature rise reduction over the 21st century. This is achieved by comparing the CESM model simulations for several SSPs under standard emission scenarios vs. scenarios based on China's carbon neutrality goals (with standard scenarios still kept for the rest of the world).

The noteworthy results of the study are, firstly, remarkable significance of impacts of Chinese carbon neutrality on temperature projections in the long term in many simulations; but, secondly, that these impacts are significant not for all SSPs, not for all time horizons, and, when it comes to geographical distributions, not for all locations.

The work is definitely significant for climate science and, more generally, for research areas aimed at supporting climate action. The reported study also has important climate policy implications.

The reported modelling work supports the conclusions and claims of the paper. The performed data analysis and interpretation are solid and reliable. This applies both to the main text of the paper and to the supplementary material.

In general, enough detail is provided in the methods for the work to be reproduced with the CESM model, and also for similar experiments to be conducted with other Earth System Models.

The paper can be recommended for publishing provided that the following minor revisions are made:

[Line 35] Ref. [2]: please double-check if the citation of Summary for Policymakers from IPCC Climate Change 2021 (yet unpublished) is appropriate already at this stage.

[Line 36] 'those quantifications based different criteria'-> 'those quantifications based on different criteria'?

[Line 53] Providing citations of papers describing the CESM model is recommended.

[Line 61] 'From the CNCN report...' – Please articulate once again that the CNCN report is Ref. [10].

[Line 115] 'Central Sarah desert' -> 'Central Sahara desert'?

[Line 116] 'Greenland sea' -> 'Greenland Sea'

The reviewer believes that the manuscript should undergo a thorough English language check.

Reviewer #2 (Remarks to the Author):

To my knowledge, there are very few studies that focus on the global temperature effect of the net zero target of a single country like China. The two perspectives that this study offers, one from global mean (Figure 1) and the other spatially explicit (Figure 2), are interesting and give some complementary aspects of the results. The study is well-structured, and the text is very clearly written. I like the concise presentation of this manuscript. This study could potentially make a valuable contribution to the literature.

However, I must say that the level of novelty that this study brings does not seem sufficient to me for publication in Nature Communications, as it stands now. In my view, a severe limitation is its

sole focus on CO₂. If I am not mistaken, China aims for GHG neutrality, not carbon neutrality (e.g. <https://www.scmp.com/news/china/science/article/3142771/chinas-2060-carbon-neutral-goal-covers-other-greenhouse-gases>). What seems more policy relevant and scientifically interesting is: how the global and regional temperature would evolve as the emissions of CO₂, CH₄, SO₂, black carbon and etc from China decline toward GHG neutrality. There are both warming and cooling climate forcers that come into play. The spatial temperature effect should be different from each short-lived climate forcer (Collins et al. 2013; Tanaka et al. 2019). Sulfate aerosol forcing is already declining due to strengthening clean air policies in China. But the warming hidden by aerosols may be unmasked at the same time (Andreae et al. 2005). A relevant and timely question is how near-term methane action, which is increasingly called for recently (CCAC (2021); <https://www.nature.com/articles/d41586-021-02287-y>), can compensate for the warming over China. This point was related to one of the headline statements of IPCC AR6 WG1: "Strong, rapid and sustained reductions in CH₄ emissions would also limit the warming effect resulting from declining aerosol pollution and would improve air quality." I would think it is necessary to incorporate non-CO₂ components in this study to make it policy relevant as well as more scientifically interesting, if this study is further considered in this journal. I have further line-by-line comments below.

Line 24

I do not understand "as quantifying historical climate responsibilities as well."

Lines 29-30

This statement uses RCP8.5 type of scenarios as a baseline. A use of such baseline is subject to ongoing debate (Hausfather and Peters 2020; Schwalm et al. 2020).

Line 39

It is perhaps "threat," instead of "threaten."

Line 42

I am not sure if this is a correct statement. The emission requirement to meet a temperature target is more complex and analyzed by Tanaka and O'Neill (2018). It largely depends on how to achieve the temperature target.

Lines 46-47

"as quantifying historical responsibilities as well" also appeared in the abstract, but I don't understand what this means here. Historical responsibilities are not analyzed in this study and another topic itself.

Line 50

As I raised at the beginning of this review report, could this be "GHG neutrality", rather than "carbon neutrality"? These two terms are explicitly defined in IPCC AR6 WG1 Annex VII. The temperature implications of these two targets are very different (Tanaka and O'Neill 2018). Please cite a source document that specifies the type of target that China aims for.

Line 52

I think this is not exactly the case. I suggest that the authors discuss a recent related study of Chen et al. (2021), which quantified the temperature effect of China's net zero, here and somewhere later (around line 83?).

Also, Duan et al. (2021), which I regard a key recent reference, should be discussed somewhere in this manuscript.

Line 86

Smaller differences, rather than less difference?

Line 87

Historical CO₂ accumulation applies equally to all scenarios. I don't think that this can mask the difference.

Line 103

Is there any reason for the contrasting cooling effect in this region?

Line 107

Like above, I wonder what the potential reasons for the spatial differences between SSP1-2.6 and SSP2-4.5 results would be. Could this be explained by spatial variability exhibited in the simulations used by this study?

Line 118

I suggest that the authors make some statements on to what extent their findings are contingent on the simulations used by this study. Could their results be very different if they have an opportunity to simulate the model in the same setting multiple times?

Line 126

As I pointed out earlier, an underlying assumption in the argument is that the authors refer to SSP3-7.0 and SSP5-8.5 as a baseline. However, these baselines could be too high, given the recent development (Cornwall 2020).

Line 127

The study did not really test the sensitivity of the results with respect to the amount and date of carbon peak and neutrality.

Figure 2

I find it difficult to identify the signs of plus (showing statistical significance) in this figure.

References

Andreae MO, Jones CD, Cox PM (2005) Strong present-day aerosol cooling implies a hot future. *Nature* 435 (7046):1187-1190

CCAC (2021) Global Methane Assessment.

Chen J, Cui H, Xu Y, Ge Q (2021) Long-term temperature and sea-level rise stabilization before and beyond 2100: Estimating the additional climate mitigation contribution from China's recent 2060 carbon neutrality pledge. *Environ Res Lett*. doi:10.1088/1748-9326/ac0cac

Collins WJ, Fry MM, Yu H, Fuglestedt JS, Shindell DT, West JJ (2013) Global and regional temperature-change potentials for near-term climate forcings. *Atmos Chem Phys* 13 (5):2471-2485. doi:10.5194/acp-13-2471-2013

Cornwall W (2020) Five years in, Paris pact still a work in progress. *Science* 370 (6523):1390-1390. doi:10.1126/science.370.6523.1390

Duan H, Zhou S, Jiang K, Bertram C, Harmsen M, Kriegler E, Vuuren DPv, Wang S, Fujimori S, Tavoni M, Ming X, Keramidas K, Iyer G, Edmonds J (2021) Assessing China's efforts to pursue the 1.5°C warming limit. *Science* 372 (6540):378-385. doi:doi:10.1126/science.aba8767

Hausfather Z, Peters GP (2020) RCP8.5 is a problematic scenario for near-term emissions. *Proceedings of the National Academy of Sciences* 117 (45):27791-27792. doi:10.1073/pnas.2017124117

Schwalm CR, Glendon S, Duffy PB (2020) RCP8.5 tracks cumulative CO₂ emissions. *Proceedings of the National Academy of Sciences* 117 (33):19656-19657. doi:10.1073/pnas.2007117117

Tanaka K, Cavalett O, Collins WJ, Cherubini F (2019) Asserting the climate benefits of the coal-to-gas shift across temporal and spatial scales. *Nature Climate Change* 9 (5):389-396. doi:10.1038/s41558-019-0457-1

Tanaka K, O'Neill BC (2018) The Paris Agreement zero-emissions goal is not always consistent with the 1.5 °C and 2 °C temperature targets. *Nature Climate Change* 8 (4):319-324.

Reviewer #3 (Remarks to the Author):

This paper explores an interesting topic, i.e. how pledges of individual countries contribute to global warming. However, the paper is not well written and has some fundamental methodological issues. It's not clear how this research contributes to the literature.

1. There are broken sentences and typos throughout the paper, making it difficult to understand. For example:

- lines 21-24, the first sentence has several grammar mistakes.
- line 36 based on different criteria, not "based different criteria"
- line 39 broad threats, not "broad threaten"
- line 49 CO2 emissions., not "CO2 emission"
- line 58 sixth phase or phase six, not "six phase of Coupled Model Intercomparison Project"
- Line 64: "This causes a difference in anthropogenic surface CO2 emissions between the default CMIP6 and CNCN scenarios". It's not clear what "this" means in this sentence.

2. It seems that the CNCN scenario is key to the results of this paper. The authors need to better explain how the CNCN scenario/report estimates China's emissions and how it differs from other studies assessing China's carbon neutrality targets. China's NDC and LTS pledges only define peaking and carbon neutrality time, but don't specify pathways. Different pathways can be taken to meet China's pledges and would result in different temperature outcomes. The authors need to consider uncertainties in China's carbon neutrality pathways.

3. One of the main findings of this paper is that "China's carbon neutrality can individually mitigate global warming by 0.48 (± 0.09) °C and 0.40 (± 0.09) °C, which accounts for 14% and 9% of average increase in global mean surface temperature over the long term (2081-2100) under scenarios of SSP3-7.0 and SSP5-8.5, respectively." This basically assumes that other countries don't take significant mitigation actions. The reality is the opposite. The paper should at least discuss emissions mitigation from the rest of World and China's contribution to global warming when all countries meet their pledges.

4. The paper should also discuss assumptions on non-CO2 GHGs, as they would affect temperature outcomes.

Dear Reviewers,

Many thanks for reviewing our manuscript. Please find our response (shown in blue text) to the individual comments. When showing changes to the text, new sentences/words are shown in *bold and italic* in this response letter.

Reviewer #1 (Remarks to the Author):

Q1 -- The paper under review provides quantitative assessment of potential contributions of carbon neutrality of Chinese economy to global and regional temperature rise reduction over the 21st century. This is achieved by comparing the CESM model simulations for several SSPs under standard emission scenarios vs. scenarios based on China's carbon neutrality goals (with standard scenarios still kept for the rest of the world).

The noteworthy results of the study are, firstly, remarkable significance of impacts of Chinese carbon neutrality on temperature projections in the long term in many simulations; but, secondly, that these impacts are significant not for all SSPs, not for all time horizons, and, when it comes to geographical distributions, not for all locations.

The work is definitely significant for climate science and, more generally, for research areas aimed at supporting climate action. The reported study also has important climate policy implications.

The reported modelling work supports the conclusions and claims of the paper. The performed data analysis and interpretation are solid and reliable. This applies both to the main text of the paper and to the supplementary material.

In general, enough detail is provided in the methods for the work to be reproduced with the CESM model, and also for similar experiments to be conducted with other Earth System Models.

The paper can be recommended for publishing provided that the following minor revisions are made:

Response: We thank the reviewer for the positive evaluations.

Q2 -- [Line 35] Ref. [2]: please double-check if the citation of Summary for Policymakers from IPCC Climate Change 2021 (yet unpublished) is appropriate already at this stage.

Response: Thanks for the comment, now the IPCC AR6 SPM can be cited from its official website: https://www.ipcc.ch/report/ar6/wg1/downloads/report/IPCC_AR6_WGI_SPM.pdf.

Q3 -- [Line 36] 'those quantifications based different criteria' -> 'those quantifications based on different criteria'?

Response: "on" has been inserted.

Q4 -- [Line 53] Providing citations of papers describing the CESM model is recommended.

Response: A paper¹ published in Journal of Advances in Modeling Earth Systems is cited.

Q5 -- [Line 61] ‘From the CNCN report...’ – Please articulate once again that the CNCN report is Ref. [10].

Response: Ref. [10] (No. Ref [13]² in the revised manuscript) was inserted.

Q6 -- [Line 115] ‘Central Sarah desert’ -> ‘Central Sahara desert’?

Response: “Sarah” was corrected to “Sahara”.

Q7 -- [Line 116] ‘Greenland sea’ -> ‘Greenland Sea’

Response: Corrected.

Q8 -- The reviewer believes that the manuscript should undergo a thorough English language check.

Response: Thanks for the suggestion and language of the manuscript has been edited by Springer Nature.

Reviewer #2 (Remarks to the Author):

Q1 -- To my knowledge, there are very few studies that focus on the global temperature effect of the net zero target of a single country like China. The two perspectives that this study offers, one from global mean (Figure 1) and the other spatially explicit (Figure 2), are interesting and give some complementary aspects of the results. The study is well-structured, and the text is very clearly written. I like the concise presentation of this manuscript. This study could potentially make a valuable contribution to the literature.

Response: We thank the reviewer for the positive evaluations.

Q2 -- However, I must say that the level of novelty that this study brings does not seem sufficient to me for publication in Nature Communications, as it stands now. In my view, a severe limitation is its sole focus on CO₂. If I am not mistaken, China aims for GHG neutrality, not carbon neutrality (e.g. <https://www.scmp.com/news/china/science/article/3142771/chinas-2060-carbon-neutral-goal-covers-other-greenhouse-gases>). What seems more policy relevant and scientifically interesting is: how the global and regional temperature would evolve as the emissions of CO₂, CH₄, SO₂, black carbon and etc from China decline toward GHG neutrality. There are both warming and cooling climate forcers that come into play. The spatial temperature effect should be different from each short-lived climate forcer (Collins et al. 2013; Tanaka et al. 2019). Sulfate aerosol forcing is already declining due to strengthening clean air policies in China. But the warming hidden by aerosols may be unmasked at the same time (Andreae et al. 2005). A relevant and timely question is how near-term methane action, which is increasingly called for recently (CCAC (2021); <https://www.nature.com/articles/d41586-021-02287-y>), can compensate for the warming over China. This point was related to one of the headline statements of IPCC AR6 WG1: “Strong, rapid

and sustained reductions in CH₄ emissions would also limit the warming effect resulting from declining aerosol pollution and would improve air quality.” I would think it is necessary to incorporate non-CO₂ components in this study to make it policy relevant as well as more scientifically interesting, if this study is further considered in this journal.

Response:

Concerning the parlance that China aims for carbon neutrality or GHG neutrality, what we learned from the official news in China is carbon neutrality, although the government will make best efforts for GHG neutrality. This statement of carbon neutrality for China is also introduced as “**China’s plan focuses only on CO₂ emissions and does not include methane or nitrous oxide**” in a “Editorials” article of Nature (Net-zero carbon pledges must be meaningful. Nature 2021, 592(8))³ and “**That’s on top of China’s commitment earlier this year to reduce its net carbon emissions to zero by 2060**” in a News article in Science (Cornwall, 2022)⁴. In addition, the co-author of the manuscript, Prof. Tianjun Zhou, who is a lead author of IPCC AR6 WGI, confirmed that China is currently aiming at carbon neutrality. This is also confirmed from a colleague in Chinese Academy of Social Sciences who is expertized in policy research on China’s carbon neutrality.

Even though, in the revised manuscript, we recognise the importance of non-CO₂ GHG effects on future global temperature and have conducted additional simulations with inclusion of CH₄ and N₂O in association with carbon neutrality for four SSPs (referred to as CNGN). We find that incorporating non-CO₂ GHGs does not make significant impacts on GMST for all SSPs over the near term, but produces significant impacts on GMST for two out of four SSSPs over the mid-term and all four SSPs over the long term (see Fig. 1 in the manuscript and corresponding descriptions in Lines 102-112) and such impacts are spatially divergent and scenario-dependent (see Fig.2 in the manuscript and corresponding descriptions in Lines 148-164).

As for SO₂, BC and other greenhouse gases, we lack confidence in model projections so do not incorporate them into the model, but we extended our discussions as “***Third, changes in atmospheric aerosols, such as short lived GHGs cooccurring with fossil fuel combustions and diverse human activities, are not taken into account in the simulations, although aerosols are reported to but have conflicting effects that range from significant reduction in temperature to a modest impact and even a net future warming effect***^{5, 6, 7, 8}” (see Lines 189-192).

I have further line-by-line comments below.

Q3 -- Line 24: I do not understand “as quantifying historical climate responsibilities as well.”

Response: Thanks. The sentence has been revised as: “***projecting mitigations of pledged carbon neutrality from individual countries to future global warming is of the same importance as previous efforts devoted to quantifying historical climate responsibilities***”. (see Lines 22-24)

Q4 -- Lines 29-30: This statement uses RCP8.5 type of scenarios as a baseline. A use of such baseline is subject to ongoing debate (Hausfather and Peters 2020; Schwalm et al. 2020).

Response: We add a sentence after the introduction of SSP5-85 “*although the SSP5-8.5 scenario is criticised as overestimating future cumulative fossil fuel and industry CO₂ emissions*” in Methods section (Lines 227-228).

Q5 -- Line 39: It is perhaps “threat,” instead of “threaten.”

Response: “threaten” has been corrected as “threats”.

Q6 -- Line 42: I am not sure if this is a correct statement. The emission requirement to meet a temperature target is more complex and analyzed by Tanaka and O'Neill (2018). It largely depends on how to achieve the temperature target.

Response: This sentence has been revised as “*reaching net zero of global CO₂ emissions in 2055 and limiting non-CO₂ greenhouse gas (GHG) emissions after 2030 are crucial mitigation strategies*”. (Lines 46-47)

Q7 -- Lines 46-47: “as quantifying historical responsibilities as well” also appeared in the abstract, but I don’t understand what this means here. Historical responsibilities are not analyzed in this study and another topic itself.

Response: What we mean here is to emphasize that projecting mitigations of pledged carbon neutrality to future global warming is as of great significance as previous efforts devoted to quantifying the historical climate responsibilities. Therefore, the sentence has been revised as “*projecting mitigations of pledged carbon neutrality from individual countries to future global warming is of the same importance as previous efforts devoted to quantifying historical climate responsibilities*”. (Lines 50-52)

Q8 -- Line 50: As I raised at the beginning of this review report, could this be “GHG neutrality”, rather than “carbon neutrality”? These two terms are explicitly defined in IPCC AR6 WG1 Annex VII. The temperature implications of these two targets are very different (Tanaka and O'Neill 2018). Please cite a source document that specifies the type of target that China aims for.

Response: As responded in Q2 comment, China is aiming at reaching carbon neutrality, not GHG neutrality at the current stage. In the revised manuscript, we supplied the simulations with inclusion of CH₄ and N₂O, and extended our discussions about uncertainties induced by other kinds of GHGs effects (see also the response for Q2).

A “Editorials” article published in *Nature*³ and another News article in *Science*⁴ are cited to specify the type of neutrality (see also the response for Q2).

Q9 -- Line 52: I think this is not exactly the case. I suggest that the authors discuss a recent related study of Chen et al. (2021), which quantified the temperature effect of China’s net zero, here and somewhere later (around line 83?).

Also, Duan et al. (2021), which I regard a key recent reference, should be discussed somewhere in this manuscript.

Response: Thanks for the comment. We have revised this sentence as “*A recent study based on a very simplified climate model reported that China’s carbon neutrality alone will contribute a 0.16-0.21 °C avoided global warming at the end of 21st century⁹. However, the magnitude of such mitigation has not yet been quantified using a fully coupled earth system model that incorporates all crucial components of the climate system*”. (Lines 56-60)

The paper of Duan et al. (2021) has also been cited for comparison as “*Under the carbon neutrality pathway², China will reduce its carbon emission by 89% in 2050, which is roughly consistent with a recent synthesis for 1.5 °C target based on multiple integrated assessment models¹⁰*” in Lines 71-73 and for discussions in Lines 184-188.

Q10 -- Line 86: Smaller differences, rather than less difference?

Response: “less” was corrected as “smaller”.

Q11 -- Line 87: Historical CO₂ accumulation applies equally to all scenarios. I don’t think that this can mask the difference.

Response: We agree with you that historical CO₂ accumulations impact future temperature, but this impact can be considered equally to all scenarios as you mentioned. We extended discussions about this as “*Finally, mitigation effects are derived by pair simulations of a single factor, CO₂ only or a combination of CO₂, CH₄ and N₂O, which essentially cannot sort out an individual country’s effects on global warming because accumulated GHGs, particularly for CO₂ with longer time residence in the atmosphere, during the historical period still apply to all future simulations and induce uncertainties*”. (see Line 192-196)

Q12 -- Line 103: Is there any reason for the contrasting cooling effect in this region?

Response: When considering CH₄ and N₂O along with CO₂ scenario, this contrasting avoided warming (to avoid misunderstanding, “avoided warming” used in the revised manuscript) over the near term for SSP2-4.5 was not observed any more (see revised Fig. 2).

Actually, even in the original submission, the contrasting cooling effect was not remarkable (see Supplementary Fig. S7). Previously incorrect statement is caused by unclear presentation of the original Fig. 2, as mentioned by the reviewer (Q17).

We apologize for this misleading in the original submission. The statement on the contrasting cooling effect has been removed from the text in the revised manuscript.

Q13 -- Line 107: Like above, I wonder what the potential reasons for the spatial differences between SSP1-2.6 and SSP2-4.5 results would be. Could this be explained by spatial variability exhibited in the simulations used by this study?

Response: A same comment as Q12. Although China’s carbon neutrality generally contributed a cooling effect to global temperature, simulated temperature also showed warming hotspots for mid-term temperature under SSP2-4.5 and SSP3-7.0 in the small region of southern Greenland (Supplementary Fig.S7). We feel hard to investigate the reason for this, but we agree with your

viewpoint that model's spatial variability may lead to significant contrasting temperature effects in some small regions.

Q14 -- Line 118: I suggest that the authors make some statements on to what extent their findings are contingent on the simulations used by this study. Could their results be very different if they have an opportunity to simulate the model in the same setting multiple times?

Response: Thanks for this suggestion. Considering other comments raised by reviewers, we have provided a "special" paragraph to discuss the uncertainties of the results from four aspects as concerned by all reviewers (Please see Lines 178-198).

Q15 -- Line 126: As I pointed out earlier, an underlying assumption in the argument is that the authors refer to SSP3-7.0 and SSP5-8.5 as a baseline. However, these baselines could be too high, given the recent development (Cornwall 2020).

Response: Probably, SSP3-7.0 and SSP5-8.5 are too high, compared to current carbon emissions. SSP3-7.0 and SSP5-8.5 depict annual CO₂ emissions in 2100 with 70 Gton and 120 Gton, respectively, vs current annual emissions of 42 Gton (Friedlingstein et al., 2020. ESSD)¹¹. From the Global Carbon Budget 2020 (Friedlingstein et al., 2020. ESSD), we know that global carbon emissions has been quintuplicated since 1960 (i.e. during the past 60 years). From now to 2100, we have 80 years to go and there are still lots of developing and undeveloped countries who rely on further emissions to boost the economic development and improve incomes. Scenario of very high carbon emissions are not always impossible to happen. More importantly, SSPs are designed to facilitate the integrated analysis of future climate impacts, vulnerabilities, adaptation, and mitigation (Riahi et al., 2017. Global Environ. Change)¹². The current research makes use of wide range of carbon emission scenarios to reveal the relative contribution of China's carbon neutrality to future global warming.

To illustrate the uncertainties of high emission scenarios, we have provided some discussions in a new paragraph, as responded in Q14.

Q16 -- Line 127: The study did not really test the sensitivity of the results with respect to the amount and date of carbon peak and neutrality.

Response: This sentence has been removed from the text in the revised manuscript.

Figure 2

Q17 -- I find it difficult to identify the signs of plus (showing statistical significance) in this figure.

Response: The Figure 2 has been re-plotted in a different style so that we hope it could be clear for now.

Reviewer #3 (Remarks to the Author):

Q1 -- This paper explores an interesting topic, i.e. how pledges of individual countries contribute to global warming. However, the paper is not well written and has some fundamental methodological issues. It's not clear how this research contributes to the literature.

Response: We thank the reviewer for the positive evaluations and all comments have been carefully considered and addressed in the revised manuscript.

The results provide a useful reference for the global stocktake, which assesses the collective progress towards the climate goals of the Paris Agreement. This is provided in the Abstract (Lines 32-34) and the last paragraph of the revised manuscript.

Q2 -- 1. There are broken sentences and typos throughout the paper, making it difficult to understand. For example:

-lines 21-24, the first sentence has several grammar mistakes.

Response: This sentence has been rewritten as *“In the context of the current opportunity in which more than 120 countries, including China and the majority of the world’s large annual emitters of anthropogenic CO₂, have pledged for carbon neutrality, projecting mitigations of pledged carbon neutrality from individual countries to future global warming is of the same importance as previous efforts devoted to quantifying historical climate responsibilities”*. (Lines 20-24)

-line 36 based on different criteria, not “based different criteria”

Response: “based” was changed to “based on”.

-line 39 broad threats, not “broad threaten”

Response: Revised accordingly.

-line 49 CO₂ emissions., not “CO₂ emission”

Response: “CO₂ emission” has been revised as “CO₂ emissions”.

- line 58 sixth phase or phase six, not “six phase of Coupled Model Intercomparison Project”

Response: “six” has been changed to “sixth”.

- Line 64: “This causes a difference in anthropogenic surface CO₂ emissions between the default CMIP6 and CNCN scenarios”. It's not clear what “this” means in this sentence.

Response: This sentence has been revised as *“Compared with the default CMIP6 scenario, the CNCN has a difference ranging from -3.70 to 18.03 GtCO₂ year⁻¹ in anthropogenic surface CO₂ (Supplementary Figs. S2-3)”*. (Lines 73-75)

Q3 -- 2. It seems that the CNCN scenario is key to the results of this paper. The authors need to better explain how the CNCN scenario/report estimates China's emissions and how it differs from other studies assessing China's carbon neutrality targets. China's NDC and LTS pledges only define peaking and carbon neutrality time, but don't specify pathways. Different pathways can be taken to meet China's pledges and would result in different temperature outcomes. The authors need to consider uncertainties in China's carbon neutrality pathways.

Response: Thanks. China's NDC and LTS pledges only define peaking and carbon neutrality time, but don't specify pathways. In the CNCN report (Ref 13 in the manuscript), the pathway is generated based on two scenarios by considering national developmental strategies and goals, ecological civilization construction to meet China's Nationally Determined Contributions (NDC) and long-term low GHG emissions development strategies (LTS) and other two scenarios for meeting 2 °C and 1.5 °C temperature target under the Paris Agreement (Ref 13).

The CNCN scenario is mainly based on carbon emissions consistent with the IPCC 1.5 °C target, but it requires further reductions in national total energy consumptions and large increases in the proportion of non-fossil energy to primary energy consumptions. The CNCN scenario also requires significant decreases in non-CO₂ GHG emissions and increases in terrestrial ecosystem carbon sinks, and large-scale implementations of carbon capture and storage (CCS) and carbon dioxide removal (CDR). In the revised manuscript, these explanations have been provided in Methods section.

Following the CNCN's carbon emissions pathway², China has to reduce its carbon emission by 89% in 2050, which is quite consistent with a recent synthesis for 1.5 °C target based on multiple integrated assessment models¹⁰ (Lines 71-73).

We agree with you that emissions pathways impact temperature outcomes, however, as mentioned above, the CNCN pathway is a comprehensively entire-sector and multi-component effort to meet carbon neutrality for China. We have extended some discussions in Lines 184-188.

Q4 -- 3. One of the main findings of this paper is that "China's carbon neutrality can individually mitigate global warming by 0.48 (±0.09) °C and 0.40 (±0.09) °C, which accounts for 14% and 9% of average increase in global mean surface temperature over the long term (2081-2100) under scenarios of SSP3-7.0 and SSP5-8.5, respectively." This basically assumes that other countries don't take significant mitigation actions. The reality is the opposite. The paper should at least discuss emissions mitigation from the rest of World and China's contribution to global warming when all countries meet their pledges.

Response: Yes, the current research assumes that other countries except China don't make any changes in carbon emissions as described in the default CMIP6 scenarios. Such treatment is opposite to the future scenario but the purpose is to project and quantify the individual contribution of carbon neutrality from China, as the currently largest carbon emitter, to global warming.

As suggested, further discussions about this is provided in Lines 178-188.

Q5 -- 4. The paper should also discuss assumptions on non-CO₂ GHGs, as they would affect temperature outcomes.

Response: Thanks for the comment, but this is a comment same as the comment (Q2) raised by Reviewer #2.

We have supplied extra simulations with inclusion of CH₄ and N₂O based on the previous carbon neutrality simulations for four SSPs and updated all figures and description throughout the text. Please see our response for Q2 of Reviewer #2.

References

1. Danabasoglu G, Lamarque JF, Bacmeister J, Bailey DA, DuVivier AK, Edwards J, *et al.* The Community Earth System Model Version 2 (CESM2). *Journal of Advances in Modeling Earth Systems* 2020, **12**(2).
2. Tsinghua University IoCCaSD. Synthesis Report on China's long term low carbon development and transmission pathways. *China Population, Resources and Development* 2021, **30**: 1-25.
3. Editorials. Net-zero carbon pledges must be meaningful. *Nature* 2021, **592**(8).
4. Cornwall W. Five years in, Paris pact still a work in progress. *Science* 2020, **370**(6523): 1390.
5. Andreae MO, Jones CD, Cox PM. Strong present-day aerosol cooling implies a hot future. *Nature* 2005, **435**(7046): 1187-1190.
6. Allen RJ, Horowitz LW, Naik V, Oshima N, O'Connor FM, Turnock S, *et al.* Significant climate benefits from near-term climate forcer mitigation in spite of aerosol reductions. *Environmental Research Letters* 2021.
7. Collins WJ, Fry MM, Yu H, Fuglestvedt JS, Shindell DT, West JJ. Global and regional temperature-change potentials for near-term climate forcers. *Atmospheric Chemistry and Physics* 2013, **13**(5): 2471-2485.
8. Fu B, Gasser T, Li B, Tao S, Ciais P, Piao S, *et al.* Short-lived climate forcers have long-term climate impacts via the carbon–climate feedback. *Nature Climate Change* 2020, **10**(9): 851-855.
9. Chen J, Cui H, Xu Y, Ge Q. Long-term temperature and sea-level rise stabilization before and beyond 2100: Estimating the additional climate mitigation contribution from China's recent 2060 carbon neutrality pledge. *Environmental Research Letters* 2021, **16**(7).
10. Duan H, Zhou S, Jiang K, Bertram C, Harmsen M, Kriegler E, *et al.* Assessing China's efforts to pursue the 1.5°C warming limit. *Science* 2021, **372**(6540): 378-385.
11. Friedlingstein P, O'Sullivan M, Jones MW, Andrew RM, Hauck J, Olsen A, *et al.* Global Carbon Budget 2020. *Earth Syst Sci Data* 2020, **12**(4): 3269-3340.

12. Riahi K, van Vuuren DP, Kriegler E, Edmonds J, O'Neill BC, Fujimori S, *et al.* The Shared Socioeconomic Pathways and their energy, land use, and greenhouse gas emissions implications: An overview. *Global Environmental Change* 2017, **42**: 153-168.

Reviewer comments, second round review

Reviewer #1 (Remarks to the Author):

The reviewer is satisfied with how his minor comments have been addressed by the authors in the revised version.

The paper can be recommended for publication.

Reviewer #2 (Remarks to the Author):

I appreciate the efforts by the authors to address my review comments, in particular the additional computation that the authors performed to explore the effect of non-CO2 GHG mitigation on the surface temperature. I have several remaining comments below.

GHG neutrality

If I understand correctly from lines 247-266, the authors assume same CO2 emission reductions for CNCN and CNGN. However, it is well known that CO2 emission pathways for carbon neutrality and GHG neutrality are different. Under the assumptions of many current IAMs, CO2 emission pathways for GHG neutrality must go negative to compensate for residual non-CO2 emissions (i.e. remaining CH4 and N2O emissions from difficult-to-abate sectors such as agriculture and livestock). In other words, negative CO2 emissions and some positive non-CO2 emissions cancel out each other (using GWP100 weighting) to achieve a GHG neutrality. See, for instance, Tanaka and O'Neill (2018, *Nature Climate Change*, <https://www.nature.com/articles/s41558-018-0097-x>), Fuglestedt et al. (2018, <https://royalsocietypublishing.org/doi/10.1098/rsta.2016.0445>), and van Soest (2019, *Nature Communications*, <https://www.nature.com/articles/s41467-021-22294-x>). Given this, I would say that the authors did not simulate a China's GHG neutral pathway. CNGN is rather an extension of CNCN with additional CH4 and N2O emission reductions accompanied by the CO2 emission reductions for CNCN. CNGN is fundamentally different from a GHG neutrality as defined in the Article 4.1 of the Paris Agreement. However, CNGN may conform to what China aims for as the authors explained to me in detail in response to my earlier comment on Q2 (which I highly appreciate).

I suggest that the authors call CNGN under a different name to avoid confusion (or at least clarify this is different from a GHG neutrality of the Paris Agreement and the definition in IPCC AR6). I further suggest that the authors discuss the relevance of CNGN to China's actual policy including non-CO2 and the differences with the GHG neutrality in the context of the Paris Agreement based on literature suggested above.

Lines 249-250

MAGICC is not an IAM, but a simple climate model. As a paper from the community using simple climate models, see Nicholls et al. (2020, *Geoscientific Model Development*, <https://gmd.copernicus.org/articles/13/5175/2020/>).

Lines 252-253

The authors wrote: "Fortunately, both CH4 and N2O concentrations in the troposphere are well correlated with cumulative emissions (R2 close to 1, Supplementary Fig. S5) for four SSPs under the default CMIP6 scenario, ..." But this statement comes rather surprise to me because I would expect that cumulative CH4 emissions are not a good predictor of CH4 concentration or forcing, while cumulative CH4 emissions can be a good predictor for N2O concentration or forcing. CH4 is a flow gas and N2O is a stock gas (see Allen et al., 2022, *npj Climate and Atmospheric Sciences*, <https://www.nature.com/articles/s41612-021-00226-2>).

Figure S7

This figure could be moved to the main text since it is central to the discussion of the manuscript, as far as the space allows.

Dear Reviewers,

Many thanks for reviewing our manuscript. Please find our response (shown in blue text) to the individual comments. When showing changes to the text, new sentences/words are shown in *bold and italic* in this response letter.

Reviewer #1 (Remarks to the Author):

The reviewer is satisfied with how his minor comments have been addressed by the authors in the revised version.

The paper can be recommended for publication.

Response: We thank the reviewer for the positive evaluations.

Reviewer #2 (Remarks to the Author):

I appreciate the efforts by the authors to address my review comments, in particular the additional computation that the authors performed to explore the effect of non-CO₂ GHG mitigation on the surface temperature. I have several remaining comments below.

Response: We thank the reviewer for the positive evaluations, and the remaining comments have been addressed below.

Q1 -- GHG neutrality

Q1.1 -- If I understand correctly from lines 247-266, the authors assume same CO₂ emission reductions for CNCN and CNGN. However, it is well known that CO₂ emission pathways for carbon neutrality and GHG neutrality are different. Under the assumptions of many current IAMs, CO₂ emission pathways for GHG neutrality must go negative to compensate for residual non-CO₂ emissions (i.e. remaining CH₄ and N₂O emissions from difficult-to-abate sectors such as agriculture and livestock). In other words, negative CO₂ emissions and some positive non-CO₂ emissions cancel out each other (using GWP₁₀₀ weighting) to achieve a GHG neutrality. See, for instance, Tanaka and O'Neill (2018, Nature Climate Change, <https://www.nature.com/articles/s41558-018-0097-x>), Fuglestvedt et al. (2018, <https://royalsocietypublishing.org/doi/10.1098/rsta.2016.0445>), and van Soest (2019, Nature Communications, <https://www.nature.com/articles/s41467-021-22294-x>).

Response: Thanks and we agree with these comments. The raised questions are addressed in Q1.3.

Q1.2 -- Given this, I would say that the authors did not simulate a China's GHG neutral pathway. CNGN is rather an extension of CNCN with additional CH₄ and N₂O emission reductions accompanied by the CO₂ emission reductions for CNCN. CNGN is fundamentally different from a GHG neutrality as defined in the Article 4.1 of the Paris Agreement. However, CNGN may

conform to what China aims for as the authors explained to me in detail in response to my earlier comment on Q2 (which I highly appreciate).

Response: Thanks. Yes, we actually simulated an extension of CNCN with additional CH₄ and N₂O emission reductions accompanied by the CO₂ emission reductions for CNCN. This is clarified in the revised manuscript. We agree with these comments and the raised questions are addressed in Q1.3.

Q1.3 -- I suggest that the authors call CNGN under a different name to avoid confusion (or at least clarify this is different from a GHG neutrality of the Paris Agreement and the definition in IPCC AR6). I further suggest that the authors discuss the relevance of CNGN to China's actual policy including non-CO₂ and the differences with the GHG neutrality in the context of the Paris Agreement based on literature suggested above.

Response: Thanks. As suggested, we renamed CNGN to CNCN_{ext} (subscript "ext" referred to "extension" as suggested by the reviewer) to avoid confusion with the definitions in IPCC or the Paris Agreement.

As suggested, we extended and re-organized relevant discussions as *"It is well known that CO₂ emission pathways for carbon neutrality and GHG neutrality are different. Carbon neutrality targets a balance between anthropogenic emissions by sources and removals by sinks of carbon, but GHG neutrality refers to all greenhouse gases, which means that additional negative CO₂ emissions and some non-CO₂ GHG emissions have to cancel out each other for GHG neutrality^{1, 2, 3}. China has delivered a series of domestic strategies and policies including abated coal consumption⁴, clear energy development^{5, 6, 7}, nationwide ecological restoration^{8, 9} and other various negative-emission technologies¹⁰ as potential countermeasures to achieve carbon neutrality by 2060. Most of these China's ongoing emission actions also contribute to reductions in some non-CO₂ GHG emissions and increases in negative CO₂ emissions, which implies that China's future emission pathway is ultimately targeting for a GHG neutrality, although carbon neutrality is currently claimed^{11, 12}."* (see Lines 197-207)

Q2 -- Lines 249-250

MAGICC is not an IAM, but a simple climate model. As a paper from the community using simple climate models, see Nicholls et al. (2020, Geoscientific Model Development, <https://gmd.copernicus.org/articles/13/5175/2020/>).

Response: Thanks. Corrected.

Q3 -- Lines 252-253

The authors wrote: "Fortunately, both CH₄ and N₂O concentrations in the troposphere are well correlated with cumulative emissions (R² close to 1, Supplementary Fig. S5) for four SSPs under the default CMIP6 scenario, ...". But this statement comes rather surprise to me because I would

expect that cumulative CH₄ emissions are not a good predictor of CH₄ concentration or forcing, while cumulative CH₄ (*should be N₂O*) emissions can be a good predictor for N₂O concentration or forcing. CH₄ is a flow gas and N₂O is a stock gas (see Allen et al., 2022, npj Climate and Atmospheric Sciences, <https://www.nature.com/articles/s41612-021-00226-2>).

Response: Thanks for this good comment and we do agree that cumulative N₂O emissions can be a good “predictor” of N₂O concentration, but this may not hold for CH₄. Actually, the Figure S5 (below) does support this point. We can see that the dependences of concentration on cumulative emission for CH₄ under different SSPs are more “third degree polynomial”, but the dependences of concentration on cumulative emission for N₂O are much “linear”. This means that we just use an empirical cubic, not linear, function to calculate CH₄ concentrations from its cumulative emissions. We guess the phrase “concentrations are well correlated with cumulative emissions” in the sentence has caused a mis-understanding. To avoid confusion, we revised the statement as *“Fortunately, dependences of both CH₄ and N₂O concentrations on their cumulative emissions in the troposphere during the period 2015-2100 can be empirically fitted by cubic functions for four SSPs under the default CMIP6 scenario (adjust R² close to 1, Supplementary Fig. S5)”*.

Figure S5. Dependence of global CH₄ (top) or N₂O (bottom) concentrations on cumulative emissions for the four SSPs under the default CMIP6 scenarios during the period 2015-2100.

Q4 -- Figure S7

This figure could be moved to the main text since it is central to the discussion of the manuscript, as far as the space allows.

Response: The Figure S7 is moved to the main text as the Figure 2, and the original Figure 2 has been numbered as Figure 3. All citations to the figures are updated. Thanks for this valued suggestion.

References

- S1. Tanaka K, O'Neill BC. The Paris Agreement zero-emissions goal is not always consistent with the 1.5 °C and 2 °C temperature targets. *Nature Climate Change* **8**, 319-324 (2018).
- S2. Fuglestedt J, *et al.* Implications of possible interpretations of 'greenhouse gas balance' in the Paris Agreement. *Philos Trans A Math Phys Eng Sci* **376**, (2018).
- S3. van Soest HL, den Elzen MGJ, van Vuuren DP. Net-zero emission targets for major emitting countries consistent with the Paris Agreement. *Nat Commun* **12**, 2140 (2021).
- S4. Cui RY, *et al.* A plant-by-plant strategy for high-ambition coal power phaseout in China. *Nat Commun* **12**, 1468 (2021).
- S5. Xing X, *et al.* Spatially explicit analysis identifies significant potential for bioenergy with carbon capture and storage in China. *Nat Commun* **12**, 3159 (2021).
- S6. Wang P, Zhang S, Pu Y, Cao S, Zhang Y. Estimation of photovoltaic power generation potential in 2020 and 2030 using land resource changes: An empirical study from China. *Energy* **219**, 119611 (2021).
- S7. Zhou S, Wang Y, Zhou Y, Clarke LE, Edmonds JA. Roles of wind and solar energy in China's power sector: Implications of intermittency constraints. *Applied Energy* **213**, 22-30 (2018).
- S8. Lu F, *et al.* Effects of national ecological restoration projects on carbon sequestration in China from 2001 to 2010. *Proc Natl Acad Sci U S A* **115**, 4039-4044 (2018).
- S9. Wang J, *et al.* Large Chinese land carbon sink estimated from atmospheric carbon dioxide data. *Nature* **586**, 720-723 (2020).
- S10. Liu Z, *et al.* Challenges and opportunities for carbon neutrality in China. *Nature Reviews Earth & Environment* **3**, 141-155 (2022).
- S11. Editorials. Net-zero carbon pledges must be meaningful. *Nature* **592**, 8 (2021).
- S12. Cornwall W. Five years in, Paris pact still a work in progress. *Science* **370**, 1390 (2020).